# RETRACTED: Potential Protective Role of Melatonin in Benign Mammary Cells Reprogrammed by Extracellular Vesicles from Malignant Cells

**DOI:** 10.3390/biomedicines11102837

**Published:** 2023-10-19

**Authors:** Caroline Procópio de Oliveira, Barbara Maria Frigieri, Heidge Fukumasu, Luiz Gustavo de Almeida Chuffa, Adriana Alonso Novais, Debora Aparecida Pires de Campos Zuccari

**Affiliations:** 1Cancer Molecular Research Laboratory (LIMC), Faculdade de Medicina de São José do Rio Preto—FAMERP, Av. Brigadeiro Faria Lima, São José do Rio Preto 15090-000, SP, Brazil; carolineprocopio_oliveira@hotmail.com (C.P.d.O.); barbara.frigieri@unesp.br (B.M.F.); 2Postgraduate Program in Health Sciences, Faculdade de Medicina de São José do Rio Preto—FAMERP, Av. Brigadeiro Faria Lima, 5416, São José do Rio Preto 15090-000, SP, Brazil; 3Institute of Biosciences, Letters and Exact Sciences (IBILCE) UNESP, São José do Rio Preto 15054-000, SP, Brazil; 4Faculty of Animal Science and Food Engineering, University of São Paulo, Pirassununga 13635-900, SP, Brazil; fukumasu@usp.br; 5Department of Structural and Functional Biology, Institute of Biosciences, São Paulo State University (UNESP), Botucatu 18618-689, SP, Brazil; luiz-gustavo.chuffa@unesp.br; 6Institute of Health Sciences (ICS), Federal University of Mato Grosso (UFMT), Sinop 78550-728, RS, Brazil; aanovais@terra.com.br

**Keywords:** breast cancer, extracellular vesicles (EVs), melatonin

## Abstract

(1) Background: Mammary neoplasms in female dogs share many similarities with the same tumor class in humans, rendering these animals a valuable preclinical model for studying novel therapies against breast cancer. The intricate role of extracellular vesicles (EVs), particularly exosomes, in breast carcinogenesis, by transferring specific proteins to recipient cells within the tumor microenvironment, underscores their significance. Melatonin, a hormone recognized for its antitumor effects, adds another layer of intrigue. (2) Methods: EVs obtained from the plasma of dogs diagnosed with mammary tumors were co cultivated with the benign epithelial lineage E-20 using DMEM. The experiment comprised four 24 h treatment groups: control, EVs, melatonin, and EVs + melatonin. A series of assays were conducted, including colony formation, proliferation, and cellular migration assessments. Furthermore, we conducted colony formation, proliferation, and cellular migration assays. We performed immunohistochemistry for proteins of the mTOR pathway, including mTOR and AKT. (3) Results: Exosomes alone significantly increased proliferation, migration, and colony formation rates and, upregulated the expression of mTOR and AKT proteins. However, when melatonin was added, a protective effect was observed. (4) Conclusions: These findings contributed to the use of melatonin to modulate EV-mediated signaling in the clinical veterinary oncology of mammary tumors.

## 1. Introduction

The canine mammary tumor is a valuable clinical model for breast cancer (BC) in women because it shares etiological, clinical, histopathological, molecular, and prognostic similarities with human disease [1,2]. The combination of the inherently shorter lifespan of dogs, spontaneous disease manifestation, and multifactorial etiology allows for improved data collection and facilitates the development of a pragmatic clinical model [3]. Furthermore, the intricate role of extracellular vesicles in the tumor microenvironment, particularly exosomes in breast carcinogenesis and metastatic tumor progression, remains to be explored [4,5]. Extracellular vesicles (EVs) are membranous structures released by various cell types that carry bioactive cargo including mRNAs, microRNAs (miRNAs), DNA, lipids, and proteins. They circulate in body fluids and can deliver cargo to neighboring or distant cells, contributing to physiological maintenance. Based on their size and biogenesis, EVs are categorized into three types: exosomes (30–120 nm), microvesicles (40–1000 nm), and apoptotic bodies (>1000 nm). Exosomes interact with target cells through receptors, endocytosis, fusion with the plasma membrane, or the release of their cargo. Those derived from tumor cells often carry disease-related molecular signals and effectors, such as mutant oncoproteins, oncogenic transcripts, microRNAs, and DNA sequences. Microvesicles are slightly larger (40–1000 nm) and are released by pinching off the plasma membrane via a direct budding process. Apoptotic bodies are the largest (greater than 1000 nm) and are synthesized during apoptosis [6].

In this context, liquid biopsy, as a reliable platform for the discovery of candidate proteins for BC biomarkers, could be used for the early diagnosis and prognosis of both species [7]. In this study, we cocultured exosomes isolated from the plasma of female dogs with mammary carcinoma with the benign epithelial E-20 cell line to confirm their influence on the phenotype of these cells. In addition, we cocultured the isolated exosomes with melatonin to assess their ability to modulate these effects. The outcome of this study highlights the impact of tumor cell-derived vesicles on the phenotype of benign cells and the potential interference of melatonin in this mechanism.

Melatonin is a neurohormonal molecule that plays a crucial role in regulating the circadian rhythm, i.e., the body’s natural cycle of sleep and wakefulness. Its chemical structure is relatively simple, comprising an indole ring and a two-carbon side chain. The chemical formula of melatonin is C_13_H_16_N_2_O_2_. Structurally, melatonin features an indole ring that consists of a six-membered structure containing five carbon atoms and one nitrogen atom. The side chain, attached to one of the nitrogen atoms in the indole ring, consists of a single methylamine molecule (CH_3_NH) followed by a two-carbon chain. Melatonin production occurs in the pineal gland of the brain and is influenced by light and darkness. During the night, when exposure to light is minimal, the pineal gland increases melatonin production, signaling to the body that it is time to relax and prepare for sleep. This regulation of the circadian rhythm is crucial for the proper functioning of the human biological clock. [8]. Its ability to inhibit tumor progression and its oncostatic effects have been demonstrated in several studies [9]. Studies conducted by our research group showed that treatment with melatonin significantly inhibited the survival, migration, and invasion of a human breast cancer cell line (MDA-MB-231) [10]. Furthermore, another study conducted by our team, using human breast cancer cell lines MDA-MB-231 (metastatic) and MCF-7 (nonmetastatic), demonstrated that melatonin exhibits oncostatic, antimetastatic, and antiangiogenic properties, in addition to its ability to modulate ROCK-1 expression through inhibition [10]. Elevated expression of this protein is associated with the promotion of growth and metastasis Notably, MDA-MB-321 cells were more sensitive to high doses and prolonged treatment with melatonin in in vitro and in vivo experiments. Furthermore, in triple-negative breast cancer cell lines (MDA-MB-468 and MDA-MB-231), the efficacy of melatonin in suppressing cell invasion was demonstrated when treated with 1 mM for 24 h, further reinforcing its role in metastasis inhibition [11].

In addition to exhibiting antioxidant, anti-inflammatory, antitumor, and antiaging activities, melatonin has recently been studied for its influence on EVs [11]. Therefore, the impact of melatonin on the prognosis of breast cancer through EVs is questioned.

## 2. Materials and Methods

### 2.1. Treatments

Melatonin was produced commercially as a powder (Sigma-Aldrich, St. Louis, MO, USA) and subsequently diluted to 1 mM using an equal mixture of absolute ethanol and 1_X_ phosphate-buffered saline (PBS). The choice of 1 mM melatonin was based on findings from prior research conducted by our research group [12].

EVs were obtained from both blood plasma and cell culture supernatant by ultracentrifugation. Subsequently, these isolated EVs were characterized with regard to particle size and concentration using nanoparticle tracking analysis (NTA), facilitated by a Nanosight device (NS300; NTA 3.1 Build 3.1.45; Enigma Business Park, Grovewood Road, Malvern WR14 1XZ, UK).

Five 30-s videos were captured using a sCMOS camera on a Level 15 camera, and the temperature was controlled at 37 °C (Table 1). In a later stage, the EVs were characterized in terms of morphology and size via transmission electron microscopy and in terms of the presence of specific proteins via Western blotting.

The canine samples were collected at partner clinics and veterinary hospitals affiliated with UFMT and UNESP, specifically at the UFMT Veterinary Hospital, Campus of Sinop, and the UNESP Veterinary Hospital Governador Laudo Natel, Campus of Jaboticabal.

### 2.2. EV Collection

To separate platelet-poor plasma, blood from female dogs was initially collected in tubes containing EDTA as an anticoagulant. The collected blood was then subjected to two rounds of centrifugation at 2500× *g* for 15 min at 23 °C. Subsequently, the plasma underwent three consecutive centrifugations in a refrigerated centrifuge set at 4 °C: first at 300× *g* for 10 min, then at 2500× *g* for 10 min, and finally at 16,500× *g* for 30 min. These steps were taken to sequentially remove cells, cellular debris, and EVs larger than 150 nm, following the outline in Coumans et al. (2017), as described in ‘Methodological Guidelines to Study Extracellular Vesicles [13].

After centrifugation, the plasma was carefully separated and aliquoted into 1.5-mL Eppendorf tubes, which were then stored at −80 °C for subsequent isolation and characterization of EVs.

The inclusion criteria for the canine cohort encompassed female dogs of any age, breed, and castration status, all of whom presented with a clinical diagnosis of mammary neoplasia. Subsequent histological and phenotypic diagnoses were established with or without evidence of metastatic disease.

### 2.3. Isolation of EVs from Cell Cultures

To obtain an enriched content in EVs smaller than 200 nm, specifically exosomes, EVs were extracted from blood plasma and cell culture supernatants through ultracentrifugation. The extracted material was subsequently filtered using a 0.22 μm pore filter (Kasvi)—Life Sciences Division Headquarters 5350 Lakeview Parkway S Drive Indianapolis, IN 46268 United States, and subjected to ultracentrifugation at 119,700× *g* (34,000 rpm) for 70 min at 4 °C using an Optima XE-90 Ultracentrifuge with a 70 Ti rotor (Beckman Coulter. Following the initial ultracentrifugation, the resulting pellet was reconstituted in phosphate-buffered saline (PBS) and underwent a second ultracentrifugation at 119,700× *g* for 70 min at 4 °C. Ultimately, the obtained pellet was suspended in 30 μL of calcium and magnesium-free PBS. Subsequently, the isolated EVs were characterized in terms of particle size and concentration through nanoparticle tracking analysis (NTA), facilitated by a Nanosight device (NS300; NTA 3.1 Build 3.1.45; Malvern). The dilution factor used during the measurements ranged from 1:500 to 1:1500 in PBS. Five 30-s videos were recorded, captured using an sCMOS camera set to Camera Level 15, and maintained at a controlled temperature of 37 °C.

### 2.4. Nanoparticle Tracking Analysis (NTA)

To assess particle size and concentration, exosomes isolated from blood serum were diluted in 50 μL of PBS devoid of calcium and magnesium, and their characteristics were measured using a Nanosight device (NS300; NTA 3.1 Build 3.1.45; Malvern). A dilution factor of 1:500 in PBS was used for the measurements. Five 30s video recordings were acquired, with images captured with a sCMOS camera at Camera Level 15, while maintaining a constant temperature of 37 °C.

### 2.5. Cell Culture

For the in vitro study, cells from the benign breast tumor cell line E-20 were graciously provided by Professor Dr. Heidge Fukumasu from the Laboratory of Comparative and Translational Oncology at the University of São Paulo (USP). Cells were cultured in high-glucose Dubelcco’s modified Eagle medium (DMEM) obtained from Gibco™, New York, NY, USA. The culture medium was further supplemented with 10% fetal bovine serum sourced from LGC Biotecnologia, São Paulo, Brazil and 1% penicillin/streptomycin solution from LGC Biotechnology, São Paulo, Brazil. Cultured cells were incubated in a controlled environment at 37 °C with 5% CO_2_. Upon reaching 80% confluence, the cells were divided into treatment groups as follows (Table 2).

### 2.6. Coculture of Exosomes in Cell Cultures

To elucidate the effects of exosomes within the extracellular environment and on neighboring cells in the microenvironment, a coculture experiment was conducted. Exosomes obtained from the serum of female dogs were cocultured with the benign mammary epithelial lineage E-20. For this purpose, E-20 cells were seeded in 96-well plates at a density of 200,000 cells per well, with each well containing 200 µL of supplemented DMEM, as detailed in the previous section. Subsequently, exosomes isolated from the serum were introduced into the culture at a concentration of 50 g/mL of DMEM (equivalent to 10 ug per well). The coculture was then maintained for 5 days at 37 °C in a humidified incubator with 5% CO_2_, following the methodology outlined by Troyer et al. (2017) [14].

### 2.7. Cocultivation of Melatonin with Exosomes in Cell Cultures

To validate the influence of melatonin on EVs, exosomes isolated from the serum of female dogs were cocultured with melatonin. For the negative control, PBS was added to only DMEM without melatonin and exosomes. Melatonin (Sigma-Aldrich, St. Louis, MO, USA) was used at a concentration of 1 mM and diluted in 50% PBS solution and 50% absolute ethanol following the guidelines reported by Borin et al. (2016) [11]. Melatonin was diluted in a hood, in the dark, and 10 min before the experiments to prevent degradation and/or oxidation.

### 2.8. Cell Proliferation Index

The cells were grown in 24-well culture plates, seeded at a concentration of 5 × 10^4^ in 300 µL of DMEM with 10% fetal bovine serum, and kept at 37 °C in a humid chamber and atmosphere with 5% CO_2_ for 24 h until they adhered to the substrate. After this period, the culture medium was replaced with culture medium without fetal bovine serum (DMEM 0%) to leave all cells in the same cellular stage. After 24 h, the culture medium was replaced according to the above-mentioned experiments. On day zero of the experiment, the medium without fetal bovine serum was replaced with the complete medium, and melatonin was added at the desired concentration. After 4 h of the experiment, cells from the first wells of the plate were typsinized and counted in a Neubauer chamber (hemocytometer). Cells from other wells were trypsinized and counted 24 h after the beginning of the experiment, which was performed in triplicate. Cells were photographed and counted via visual inspection.

### 2.9. Cell Migration Assay

The migration assay was performed in a Boyden chamber according to Galeti et al. (2021) [15]. The assay was performed in Costar™ 24-well plates (Corning, New York, NY, USA) utilizing Transwell™ inserts (Corning, New York, NY, USA) equipped with 8.0 µm polycarbonate membranes and a 6.5-mm diameter to evaluate the migratory capacity of the cells. Initially, 1 × 10^6^ cells were transferred to the wells (six wells each). After 24 h, the cells were treated as indicated in Table 2. At the end of the treatment period, the cells were trypsinized and counted in a Neubauer chamber to verify the number of viable cells. Following the manufacturer’s instructions, 500 µL of culture medium supplemented with 10% fetal bovine serum (FBS) (chemotoxic agent) was added to the bottom of 24-well plate wells. The inserts were then positioned in each of these wells with the supplemented culture medium, and 300 µL of culture medium without fetal bovine serum was added inside the inserts containing 1 × 10^5^ of the previously treated cells mentioned in the previous paragraph. After 24 h of incubation at 37 °C with 5% CO_2_, the culture medium of the inserts was removed and carefully absorbed using a cotton swab. The supplemented culture medium on the plate was also removed. The cells contained in the inserts were then transferred to other wells of the same plate containing 500 µL of paraformaldehyde at 4% for 15 min at room temperature. Subsequently, the inserts were stained with violet crystals for 5 min to detect migratory cells and washed in autoclaved distilled water. The surplus was then removed with the aid of a cotton swab. Visualization of the migratory cells and acquisition of the images were performed with an optical microscope with a 10× objective, placing the inverted insert on a glass slide. All experiments were performed in duplicate, and the cell count was performed using Image J^®^ software (version 1.53e, ©National Institutes of Health, Bethesda, MD, USA) [16].

### 2.10. Tumor Spheroid Formation Assay

To assess colony formation, cells were cultured in 6-well plates containing 1600 cells per well with 1000 µL of 10% culture medium. After 12 h of culture, the medium was changed, and the cells were treated with the treatment groups described in Table 2. Every two days, the medium and treatments were replaced. After 14 days of culture, the medium was removed, and the cells were washed with PBS, fixed in methanol for 15 min, and stained with 0.1% crystal violet for 10 min. Finally, the plates were washed, and colonies were photographed and counted via visual inspection.

### 2.11. Immunocytochemistry

After cell culture and treatment exposure, immunocytochemistry was performed to analyze the expression of proteins p-AKT (1:200; ab81283, Abcam, Waltham, MA, USA) and mTOR (1:200; ab84400, Abcam) using a REVEAL-Biotin-Free Polyvalent DAB kit (Bioscience, California, USA) according to Marques [17]. First, 6 × 10^4^ cells were transferred from breast tumor lines to a slide with coupled silicone in 200 µL of high glucose DMEM culture medium (Gibco™, New York, NY, USA). Then, the culture medium was removed and the treatments were applied as previously described for 24 h. Briefly, following treatment, the cells were washed with 1_X_ PBS and fixed with paraformaldehyde 4%. Then, the cells were permeabilized with 1_X_ PBS/Triton X-100 at 0.5% (Sigma-Aldrich, Missouri, USA) for 10 min. Peroxidase activity was blocked for 15 min, and protein block was applied and incubated for 10 min. The dilution of the specific primary antibodies was performed according to Table 3 in 1_X_ PBS solution and bovine serum albumin at 5%. The specific primary antibodies were incubated at 4 °C overnight. The complement and conjugate HRPs were applied, followed by the DAB chromogenic substrate, which was diluted in DAB diluent buffer (1:50). Finally, the cells were stained with Harris hematoxylin. The assembly of the slides was performed in glycerol, which was then sealed. All immunoreactions were accompanied by a negative control (no addition of the primary antibody). The slides were observed under a 40_x_ objective microscope (Olympus BX53) and analyzed via optical densitometry. The analysis was performed in triplicate, and protein expression was quantified using Image J^®^ (version 1.53e, ©National Institutes of Health, USA) [16].

### 2.12. Statistical Analysis

Experiments were performed in duplicate or triplicate, and values are expressed as ± standard error of the mean (S.E.M). To compare more than two parameters, analysis of variance (ANOVA) was used, followed by Tukey’s test. Values of *p* < 0.05 were considered significant, and all statistical analyses were performed using GraphPad Prism 8.0 software (GraphPad Software).

## 3. Results

### 3.1. Characterization of the Extracellular Vesicle Profile

Western blotting analysis of the presence of ALIX, CD63, calnexin and GRP78 in EVs obtained by ultracentrifugation, as well as total protein staining showing the purification of the main serum proteins in the different preparations enriched with EVs (Figure 1).

### 3.2. Detection and Characterization of Extracellular Vesicles

The presence, morphology and purity of EVs isolates were evaluated by transmission electron microscopy (TEM), exhibiting typical bilayer EVs structures, allowing visualization of the central characteristic of exosomes (Figure 2).

### 3.3. Combination of Melatonin and EVs Attenuates Mammary Cell Migration

After 24 h, cell migration was performed using the E-20 strain (Figure 3). Exposure to EVs did not cause significant differences compared with the control group (*p* > 0.05). On the other hand, melatonin alone (1 mM) and EVs previously treated with melatonin showed a reduction in cell migration compared with the control and EV groups (*p* < 0.05).

### 3.4. Cell Proliferation Decreased via the Application of a Combination of Melatonin and EVs

We observed an increase in cell proliferation of the E-20 lineage (Figure 4), which was exposed to EVs only for 24 h, when compared with that of the control group (*p* < 0.05). In melatonin-treated cells (1 mM), it was possible to observe a reduction in cell proliferation compared with that in the control and EV groups (*p* < 0.05). The combination of melatonin (1 mM) with EVs produced a more evident reduction in cell proliferation compared with the control and EV groups (*p* < 0.05).

### 3.5. Formation of Tumor Spheroids Reduced via Combination of Melatonin and EVs

In the benign cell line E-20 treated with melatonin (Figure 5), it was possible to observe that there was a reduction in colony formation compared with that of the control group after 24 h and when compared with that of EVs (*p* < 0.05). The EV group showed no significant differences when compared with the control group (*p* > 0.05). A greater significant reduction was observed in colony formation using the combination of melatonin and EVs compared with that in the control group (*p* < 0.05).

### 3.6. mTOR Was Reduced after Melatonin Treatment

Immunocytochemical labeling of mTOR was observed in the cytoplasm of E-20 cells (Figure 6). Benign cells exposed to EVs showed no significant differences compared with the control group (*p* > 0.05). There was evidence of decreased mTOR reactions in the cells treated with melatonin and after treatment with a combination of melatonin and EVs compared with the control (*p* < 0.05) after 24 h at a concentration of 1 mM. However, the reduction remained at the same intensity in cells treated with melatonin and EVs.

### 3.7. AKT Was Reduced after EV Exposure in the Presence of Melatonin

AKT immunoreactions were mainly located in the cytoplasm of the E-20 cell line (Figure 7). Groups treated with EVs showed no significant differences compared with the control group (*p* > 0.05). There was evidence of a decrease in cells treated with melatonin compared with the control within 24 h at a concentration of 1 mM (*p* < 0.05) and compared with EVs. However, the decrease was more evident in cells treated with melatonin and EVs (*p* < 0.05).

## 4. Discussion

Few studies have explored the potential protective role of melatonin against EVs. Therefore, our paper represents an unprecedented contribution to the literature, while supporting other studies that demonstrate the therapeutic and protective potential of melatonin in mammary cancer. Currently, despite advances for treating breast cancer, this disease is highly lethal in women and dogs; although there are promising scientific advances, new therapeutic strategies are needed. Our results showed the therapeutic ability of melatonin to control tumor growth and the metastatic process by targeting early migration, colony formation in breast cancer epithelial cell lines, and modulation of proteins belonging to the PI3K/AKT/mTOR pathway. Several studies have reported and confirmed the oncostatic effects of melatonin, including the induction of apoptosis, spontaneous cell excitation, and metastasis, mediated by different disorders of action [18]. Research has shown that melatonin can inactivate the PI3K/AKT pathway in certain cells and tissues. Melatonin-induced inactivation of PI3K/AKT can trigger a cascade of intracellular events that underlie cell survival and impact the response to inflammation and other biological processes [19]. Melatonin inactivation of the PI3K/AKT pathway is a mechanism that has been consistently observed in numerous studies [20,21]. In our own research, we obtained results showing significantly increased expression of proteins related to this pathway, further confirming this effect. To provide a brief explanation, melatonin exerts its action by modulating key molecules within the PI3K/AKT pathway, ultimately leading to its inactivation. This consistent evidence across multiple studies, including our own, underscores the robustness of melatonin’s impact on this pathway.

We showed that at pharmacological concentrations (1 mM), melatonin decreased the viability of E-20 benign epithelial cells after 24 h of treatment. Melatonin exerts antiproliferative and apoptotic effects on breast cancer cells through various disruptions. It has been demonstrated that at a concentration of 1 mM, melatonin is a pharmacological concentration capable of producing anticancer effects [22]. Likewise, previous studies [10,11] have observed that melatonin at a concentration of 1 mM can significantly decrease the proliferation of MDA-MB-231 and MCF-7 cells, inhibiting tumor cell viability and invasion.

The immense diagnostic and therapeutic potential of exosomes is due to their rich protein content and their role in the transmission of such information between cells [23]. In addition, as the material packaged inside exosomes originates directly from the mother cell, the analysis of this cargo may allow instant information about the state of the host cell to be obtained in a much simpler way than when using conventional physical biopsy, making this a possible type of analysis as a routine diagnostic platform [23]. According to our findings, EVs derived from the serum of patients with tumors facilitate the formation of an expected microenvironment awaiting metastatic tumor cells [7]. When captured by nonmalignant receptor cells, these findings provide the origins for cell transformation and phenotypic reprograming cross the tumor microenvironment, and finally result in malignant cell transformation [7]. It is important to emphasize that the interaction between malignant extracellular surfaces and epithelial cells of benign breast tumors is a complex and multifactorial phenomenon that is still being widely investigated. Understanding these processes can provide important information about tumor progression and the spread of breast cancer.

When epithelial cells from benign breast tumors are exposed to melatonin-treated extracellular vesicles, a synergistic effect is observed, which results in additional benefits. This means that the combination of melatonin-treated vesicles and benign cells has a greater effect than the sum of the individual effects of each component. This synergistic effect can be manifested by further inhibition of cell proliferation, reduced migration, and suppression of colony formation. Melatonin can modulate intracellular signaling and influence gene expression, resulting in a more favorable cellular response [24].

The mechanisms of action of melatonin and EVs differ from each other, which allows consideration of the possibility of synergism between the two. In this sense, we propose a synergistic action of melatonin and EVs of malignant tumors in the studied benign epithelial cells. In summary, our current results are informative and strongly support the use of melatonin to modulate EV-mediated signaling in the clinical oncology of breast tumors. Furthermore, these findings may contribute significantly to advances in therapeutic approaches in the field of veterinary oncology.

## 5. Conclusions

Malignant tumor cells, regardless of whether they are treated with melatonin or not, release EVs capable of influencing tumor growth and migration. Melatonin modulates EVs from malignant cells by reducing their aggressive potential while amplifying their beneficial role in reducing mTOR signaling. This potential action of melatonin may represent a viable and effective option for the treatment of malignant tumors in female dogs. The results obtained strengthen the idea of using melatonin to control EV messengers in clinical oncology practice, especially in cases of breast tumors.

It is crucial to highlight that the results obtained in in vitro assays may not be directly extrapolated to the in vivo environment and clinical practice. Additional studies, such as in vivo experiments and clinical trials, are needed to validate and fully understand the therapeutic potential of melatonin and EVs for treating malignant tumors in female dogs. These more comprehensive investigations are essential to establish the efficacy and safety of these therapeutic approaches prior to their clinical application.

## Figures and Tables

**Figure 1 biomedicines-11-02837-f001:** (**a**) Western blotting (WB) for the ALIX protein displayed an increased expression of this protein in isolated vesicles compared with that in isolated cells (cell line E-20); (**b**) CD63 immunoblot demonstrated strong expression of this protein in isolated vesicles compared with that in isolated cells (E-20 cell line); (**c**) calnexin expression was tested with immunoblotting and was detected only in cell isolates, confirming the purity of the isolated vesicles; (**d**) GRP78, another specific cellular protein, was also detected only in the isolated cells, further validating the purity of the isolated vesicles.

**Figure 2 biomedicines-11-02837-f002:** Identification of exosome spherical vesicular structures by transmission electron microscopy, characterizing extracellular vesicles (exosomes) isolated from the plasma of female dogs.

**Figure 3 biomedicines-11-02837-f003:** Cell migration following the combination of melatonin and EVs. The graphs represent the cell migration of the E-20 strain treated with melatonin and EVs for 24 hat a concentration of 1 mM (scale bar: 50 μm). Each column represents the mean + standard error of duplicates. Significant values (** *p* < 0.01, *** *p* < 0.001) and nonsignificant values (ns) were obtained using ANOVA followed by Tukey’s test.

**Figure 4 biomedicines-11-02837-f004:** Cell proliferation was achieved through the synergistic action of melatonin and EVs. Cell proliferation of the E-20 strain treated with melatonin and EVs for 24 hat a concentration of 1 mM. Each column corresponds to the mean + standard error of the duplicates. Significant values (** *p* < 0.01, *** *p* < 0.001, **** *p* < 0.0001) were analyzed using ANOVA followed by Tukey’s test.

**Figure 5 biomedicines-11-02837-f005:** Formation of tumor spheroids following application of a combination of melatonin and EVs. The graphs represent the number of colonies formed in the E-20 strain treated with melatonin and EVs for 24 hat a concentration of 1 mM. Each column corresponds to the mean + standard error of the duplicates. Significant values (** *p* < 0.01, *** *p* < 0.001) and nonsignificant values (ns) were analyzed using ANOVA followed by Tukey’s test.

**Figure 6 biomedicines-11-02837-f006:** Cytoplasmic labeling in the E-20 cell line. The decrease in cells was more evident when treated with melatonin for 24 h, using a concentration of 1 mM (scale bar: 50 μm). Each column corresponds to the mean + standard error of triplicates. Significant values (** *p* < 0.01, *** *p* < 0.001) and non-significant values (ns) were determined using ANOVA followed by Tukey’s test.

**Figure 7 biomedicines-11-02837-f007:** Cytoplasmic labeling in the E-20 cell line. The decrease in cells was more evident when treated with EVs and melatonin for 24 hat a concentration of 1 mM (scale bar: 50 μm). Each column corresponds to the mean + standard error of triplicates. Significant values (* *p* < 0.05, ** *p* < 0.01, *** *p* < 0.001, **** *p* < 0.0001) and nonsignificant values (ns) were determined using ANOVA followed by Tukey’s test.

**Table 1 biomedicines-11-02837-t001:** Results of concentration (particles/frame) and size (mode) of EVs isolated from the plasma of female dogs from the other carcinoma group (n = 10).

EVs	Concentration (frame/mL) 1ª Collect	Concentration (frame/mL) 2ª Collect	Size (Mode/nm)1ª Collect	Size (Mode/nm)2ª Collect
3a/3b	8.71 × 10^09^ ± 6.11 × 10^0^	1.12 × 10^09^ ± 3.18 × 10^07^	169.7 ± 6.8	127.3 ± 4.1
5a/5b	7.50 × 10^09^ ± 4.49 × 10^08^	3.51 × 10^09^ ± 3.19 × 10^08^	157.1 ± 5.2	152.4 ± 8.6
6a/6b	4.25 × 10^09^ ± 7.00 × 10^08^	1.81 × 10^09^ ± 2.39 × 10^08^	157.1 ± 6.2	182.4 ± 23.1
9a/9b	5.25 × 10^09^ ± 1.33 × 10^08^	5.46 × 10^09^ ± 3.13 × 10^08^	171.8 ± 4.7	133.4 ± 4.7
15a/15b	2.81 × 10^09^ ± 1.19 × 10^08^	6.62 × 10^09^ ± 5.67 × 10^08^	158.6 ± 11.5	117.0 ± 2.3
16a/16b	8.31 × 10^09^ ± 6.55 × 10^08^	10.6 × 10^09^ ± 9.91 × 10^08^	177.3 ± 4.3	146.3 ± 8.6
18a/18b	11.6 × 10^09^ ± 5.65 × 10^08^	3.42 × 10^09^ ± 1.13 × 10^08^	153.5 ± 14.5	146.4 ± 8.9
19a/19b	8.61 × 10^09^ ± 5.02 × 10^08^	4.55 × 10^09^ ± 2.74 × 10^08^	146.7 ± 7.8	155.8 ± 5.8
24a/24b	3.59 × 10^09^ ± 1.11 × 10^08^	3.37 × 10^09^ ± 1.34 × 10^08^	163.3 ± 9.0	134.1 ± 6.0
25a/25b	2.99 × 10^09^ ± 7.67 × 10^08^	1.38 × 10^09^ ± 5.57 × 10^07^	148.2 ± 2.7	147.9 ± 17.7

**Table 2 biomedicines-11-02837-t002:** Treatment groups.

Group	Treatment
Group 1	Negative control (DMEM culture)
Group 2	EVs isolated with E-20 cell line
Group 3	1 mM melatonin with E-20 cell line
Group 4	1 mM melatonin together with EVs isolated from E-20 cell line

**Table 3 biomedicines-11-02837-t003:** Antibody dilution.

Specific Antibody	Description	Dilution
mTOR	Anti-mTOR (phospho S2448) antibody (ab84400)	1:200
AKT	Anti-AKT1 (phospho S473) antibody (ab81283)	1:200

## Data Availability

Not applicable.

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
