# Peer review of "Potential Protective Role of Melatonin in Benign Mammary Cells Reprogrammed by Extracellular Vesicles from Malignant Cells"

_biomedicines, 2023, doi:10.3390/biomedicines11102837_

Round 1

Reviewer 1 Report

Procópio et al, applied the concept of melatonin as ant-tumor for the cancer produced by EVs. The idea looks brilliant, but introduction and methods are very confusing, that needs to be rewritten. I have few concerns, that needs to be incorporated.   

1.     There is a need to rewrite the methodology section of the abstract, because it is a bit difficult to understand, and what does “s” mean in "EVs s sourced" here?

2.     In the introduction part, not enough information has been provided about extracellular vesicles, which should be sufficient, such as what EVs are, their types, characteristics, and their role in cancer. It is necessary to write this in a concise manner. I would suggest to cite the recent paper Karn et al., 2022 (Extracellular Vesicle-Based Therapy for COVID-19: Promises, Challenges and Future Prospects) https://www.mdpi.com/2227-9059/9/10/1373.  

3.     In the methodology section “2.2. EV collection

Line 70-71,

For the separation of platelet-poor plasma, initially the blood was collected in tubes with EDTA anticoagulant and centrifuged twice consecutively at a speed of 2,500xg at 71 23˚C for 15 minutes.

Is blood from human patients or from female dog?

line 72-75

The centrifugation in a centrifuge refrigerated at 4˚C, at speeds of 300xg for 10 minutes, 2,500xg for 10 minutes and 16,500xg for 30 minutes, with the aim of removing cells, cellular debris and EVs greater than 150 nm, respectively. All procedure was performed according to Coumans et al. (2017), [12] in “Methodological Guidelines to Study Extracellular Vesicles”.

How did authors confirm the removal of EVs greater than 150nm at this centrifugation speed. Please clarify.

4.     In the methodology section “2.3. Isolation of EVs from Cell Cultures

NTA is not a confirmatory method for the detection of number and size of of EVs. Some dirts comes in the sample is also visible in Nanosight machine. I would suggest to do western blot using CD63 and CD9 and CD81 (markers of exosomes). For the confirmation of size, electron microscopy is the best way to confirm the EVs size.

5.     In the manuscript, it has been mentioned that 'Melatonin is capable of inhibiting tumor cell invasion and tumor growth in vivo and in vitro.' Here, it would be appropriate to provide specific examples of tumors that have been treated with Melatonin. Please elaborate it in details with recent references.

6.     There is a lack of information related to the topic in the overall introduction part. It needs to be supplemented with additional details. Please revise it accordingly.

7.     Please clarify whether this is about TNBC or NMTN for triple negative breast cancer in line 63 where it says, 'The collections were performed in the group of patients with triple-negative breast cancer (NMTN).

8.     Please remove unnecessary characters, such as 'u', in line 65.

9.     Here, two words have been used for a dog: 1. female dog and 2. Bitches; it would be better to use either of these words.

10.  Melatonin is inactivating the PI3K/AKT pathway through what action mechanism; a little explanation is needed. 

Extensive English editing is required throughout the entire manuscript. 

Reviewer 2 Report

In the manuscript entitled “Potential protective role of melatonin in benign mammary cells reprogrammed by extracellular vesicles from malignant cells”, the authors investigated the effect of melatonin and EVs on the proliferation, migration, and colony formation of breast cancer cells. The experiments were poorly designed and the description is quite confusing throughout the study. The conclusions were based on one single cell line and images with low resolution. Therefore, the current manuscript is not suitable for publication in Biomedicines.

1. Only one canine mammary cancer cell line E-20 was used for the entire study. In addition, considering the species, E-20 is not widely used nor a well-recognized model in breast cancer studies. To ensure reproducibility and scientific rigors, at least human breast cancer cell lines should be used to validate the major conclusion.

2. In Table 1, the authors stated that EVs used in groups #2 and #4 were from E-20 cells. However, it was mentioned that EVs were sourced from the plasma of dogs with mammary tumors in the abstract and methods. Please explain the origin of EVs.

3. The authors did not provide any data from Nanoparticle Tracking Analysis (NTA). Besides, according to the guidelines of ISEV, EVs need to be further characterized by western blot analysis and TEM microscopy.

4. No scale bars were included in all the figures.

5. The images are poor in quality, making it hard to believe the statistical analysis, especially Figs 4 and 5.

6. The authors stated the dilution of the specific primary antibodies was prepared according to Table 2. However, table 2 is absent.

Author Response

"Please, see the attachment."

Reviewer 3 Report

The submitted work focuses on the potential protective role of melatonin in relation to extracellular vesicles. The study is of practical importance as it may turn out that the list of possible applications of melatonin can potentially be extended in the (far) future. Also, the model has been chosen correctly and the style of presentation is acceptable. However, while the study is interesting as the well known melatonin seems to have other then already described beneficial properties, the study requires revision. The corrections required are both minor (editorial, vocabulary, etc.) as well as major ones (statistical, methodological). The list of them is presented below.

Line 3, please remove     "  

In the introduction the structure of melatonin should be presented

Line 52, here, the aim of the study should be clearly stated

Line 57 and other places too, why is Melatonin with capital “M”?

Line 65, what is a “U Federal”?

Lines 63-68, the appropriate permissions (i.e. ethical) are not listed here neither at the end of the manuscript, this must be updated

Lines 71, 73, etc. this is not a speed but acceleration

Table 1, an editorial mistake, height of the rows are not the same

Lines 120, 134, 156, etc. it should be “CO2

Line 134, “4” should be with superscript

Lines 166, 195, what version of the software? A references is also needed.

Line 186, Table 2 is missing….

Line 305 and other places too, it should be “mM” and not “Mm”

VERY IMPORTANT ONE: The statistical analysis has not been described at all in the Materials and Methods section. This must be updated with all the necessary information (i.e. software, tests applied, etc.)

Another important one: The Authors have used melatonin only at one concentration. Why this particular value has been chosen? Why the other concentrations have not been tested?

Author Response

"Please, see the attachment."

Round 2

Reviewer 1 Report

The authors addressed my comments. I would recommend it for publication.   

Reviewer 2 Report

Unfortunately, the authors did not address the major concern regarding scientific rigor and reproducibility. All the conclusions are based on one single in vitro model E-20 cell line, which is not well characterized by researchers in the breast cancer field.

Reviewer 3 Report

The Authors have provided suitable answers to my questions and made the requested changes in the manuscript. This version can be accepted.